# A person-centered approach to cognitive performance analysis in primary school children: Comparisons through self-organizing maps

Sergio Montalt-García[1,2☯], Isaac Estevan[1,2☯], Israel Villarrasa-Sapiña[1,3,4☯], Xavier García-Massó[1,2,4☯] *

1 Physical Activity and Health Promotion Research Group (AFIPS), University of Valencia, València, Spain, 2 Department of Teaching of Physical Education, Arts, and Music, University of Valencia, València, Spain, 3 Department of Physical Education and Sports, University of Valencia, València, Spain, 4 Human Movement Analysis Group (HuMAG), University of Valencia, València, Spain

☯ These authors contributed equally to this work.
* xavier.garcia@uv.es

## Abstract

The objective of this study was to identify distinct student profiles based on physical, psychological, and social characteristics, and examine their impact on cognitive performance. A total of 194 children participated in this cross-sectional design study (mean age = 10.61 years, SD = 0.45; 48.96% girls). The study included participants from diverse racial backgrounds. Using Self-Organizing Maps, an unsupervised neural network clustering technique, six distinct profiles were identified. These profiles revealed significant effects in daily physical activity, self-reported physical, social, and psychological factors, and physical performance. Profiles characterized by higher physical activity levels and positive social and psychological factors were associated with better cognitive performance, in contrast to profiles with lower levels in these domains. These findings suggest that students' cognitive outcomes may be linked to their physical, psychological, and social characteristics, which interact to shape cognitive functioning. The recognition of the diversity of student profiles in specific educational settings may facilitate the design of more targeted programs that address individual needs and strengths, thereby enhancing their development in these domains within similar educational contexts.

## Introduction

The comprehensive development of children is a crucial area of research for promoting social health and well-being, with cognitive performance (CP) being a central aspect [1]. The concept of CP involves the capacity to process information, exhibit intelligence and reasoning skills, as well as developing language and memory. These cognitive functions are typically measured through both subjective assessments (i.e. assessed by self-reports or reports from others such as academic achievement or academic self-perception) and objective assessments (i.e. assessed by standardized cognitive tests such as the Stroop, Digit Span tests or

**Data availability statement:** The data that support the findings of this study are openly available in Zenodo at http://doi.org/10.5281/zenodo.10527912. Documentation of the code is provided as Supporting information (S1 File).

**Funding:** This work was supported by the Spanish National Research Agency (URL: https://www.aei.gob.es/en) under Grant [PID2020-115075RA-I00 by the MCIN/AEI/10.13039/50110001]; and the Conselleria d'Innovació, Universitats, Ciència i Societat Digital (URL: https://avant.gva.es/va/cons-in-novacio) under Grant [AICO/2022/185]; and SM-G's pre-doctoral contract is funded by the Conselleria d'Innovació, Universitats, Ciència i Societat Digital (URL: https://avant.gva.es/va/cons-innovacio) under Grant [CIACIF/2021/172]; and the postdoctoral contract of IV-S is funded by the Universitat de València (URL: https://www.uv.es/) under Grant [CPI-21-518]. The sponsors or funders had no involvement in the study design, data collection and analysis, decision to publish, or preparation of the manuscript.

**Competing interests:** The authors have declared that no competing interests exist.

math fluency) [2]. Fostering and stimulating CP from early childhood can be beneficial for their well-being and mental health [1,3]. Formal education, participation in sports and other experiences in social and cultural settings contribute significantly to individuals' progress in CP [3–5]. However, not all individuals have access to environments that raise these processes [6]. In a constantly changing sociocultural context, it is important to provide a wide range of opportunities for the development of young children's cognitive processes. Achieving this requires a comprehensive understanding of the foundational factors that influence CP in childhood. Previous research has demonstrated that CP is associated with a multitude of factors, including genetics, age, gender, area of residence, socioeconomic status and lifestyle [7,8].

Physical activity (PA) participation is one factor extensively studied in sports science over the last decades for its influence on CP. Several physical domain factors have been identified as playing a particularly essential role in the influence of CP. Among these factors, PA practice [9–11], physical fitness [12,13], and motor competence [12,14,15] emerge as significant elements that warrant consideration. Despite some inconsistencies in the existing evidence, PA has been repeatedly identified as a significant contributing factor to CP development [16,17]. Physical fitness, for its part, is closely linked to PA, with components like cardiorespiratory fitness [14,16] and speed-agility [9,12] consistently associated with enhanced CP. Motor competence also shows emerging evidence of its relationship with CP [12,14,15]. A systematic review by van der Fels et al. [18] revealed a moderate-to-strong association between fine motor skills and visual processing, while bilateral body coordination showed a weak-to-moderate correlation with fluid intelligence. Conversely, Bao et al. [19] found low associations between gross motor skills components —such as locomotion, object control, and stability— and CP, suggesting that including neuropsychological factors in future studies could provide a deeper understanding of these relationships. However, the findings from experimental studies, particularly those conducted in school settings, remain inconclusive regarding the impact of increasing PA practice on CP. For instance, while Zeng et al. [20] found that increased PA had a positive impact on executive functioning, language, and academic achievement in four of five randomized controlled trials, Sember et al. [17] reported minimal improvements in academic performance from PA interventions that lasted a minimum six weeks. These disparate findings suggest that additional factors may influence in this relationship. In this case, the potential moderating effects of PA intensity and instructor on the effects of PA on CP were identified [17]. Meanwhile, Esteban-Cornejo et al. proposed that the type of activity and certain psychological factors, such as self-esteem or depression, could also act as mediators in this association [10]. A holistic approach related to PA beyond the physical domain that considers also psychological and social variables appears necessary to gain clearer insights into the connections between PA and CP.

In this regard, different factors from the psychological domain associated with PA, appear to exert a direct or indirect influence on CP. Motivation [4], satisfaction of basic psychological needs [21] and self-perceptions [22], such as perceived competence [23], have been identified as potential elements influencing CP development and have recently been investigated. Positive autonomous motivation —including intrinsic motivation, integrated regulation, and identified regulation— has been associated to regular PA practice [24], which may indirectly enhance children's CP by increasing PA levels [25]. However, no studies have explored the direct impact of PA practice motivation on CP. Additionally, Reeve and Lee [26] examined the neural basis of basic psychological needs within self-determination theory, finding that satisfying these needs positively correlates with activation in brain regions associated with CP. Similarly, Wang et al. [27] showed that in school settings, the satisfaction of basic needs, academic performance, and behavioral engagement are mutually reinforcing, suggesting a complex and dynamic system. Additionally, Zhen et al. [28] found that competence and relatedness (but

not autonomy) directly predict learning engagement, thereby enhancing academic achievement. Lastly, research shows that self-perceptions can play a pivotal role in both PA practice [23] (i.e. physical self-concept and perceived motor competence) and CP development [22]. In this case, age may play a crucial role in this association, as research has shown that children tend to perceive themselves more realistically as their capacity for self-perception matures [23] due to greater CP development. However, the mechanisms underlying this relationship remain unclear and may vary significantly between individuals [10,22,29].

Furthermore, given that PA is inherently a social experience, particularly in group settings or team-based activities, it is crucial to examine how social factors related to PA practice may influence CP. Variables such as peer interactions, social support, and group dynamics could significantly contribute to shaping the relationship between PA and CP [6,30,31]. Hu et al. [30] analyzed the social factors in PA participation from the perspective of the social-ecological model and highlighted interpersonal, organizational, community and political factors impacting CP. At the interpersonal and organizational levels, support from friends, parents and teachers emerged as positive predictors of students' participation in PA, alongside school interventions. At the community and policy level, access to safe facilities and safe neighborhoods were key factors in promoting PA among children [30]. Thus, these social factors may contribute directly to PA practice in childhood and throughout life, and indirectly to CP. Evidence also suggests that certain social elements, like school identity and enjoyment, can positively impact CP development and overall well-being [31,32]. For instance, O'Malley et al. [31] identified school climate and sense of identification as key predictors of academic achievement in numeracy and writing.

Understanding that the physical, psychological, and social domains not only impact PA and CP but also interact with one another is essential. Studying them in isolation could provide limited insights, as extreme scores in one factor may positively or negatively influence another. To date, no studies have considered the combined impact of a wide range of physical, psychological, and social variables on CP. This limitation may stem from traditional statistical methods becoming less effective as more independent variables are included. A variable-centered approach, compared to a person-centered method based on multiple input variables, may yield limited evidence when addressing this research gap. Indeed, a previous study employing a person-centered approach using self-organizing maps and clustering to analyze physical and cognitive variables, identified distinct student profiles in dual-task performance with varying CP levels [33]. These profiles helped to identify specific needs and strengths among students, offering a clearer understanding of how various factors influence their overall development. This highlights that children differ in their performance and responses, and some variable relationships may not be consistent across all profiles. A detailed analysis of how these elements affect CP could provide an overall perspective, enabling interventions and policies to be tailored to specific needs and challenges. The present study aimed to comprehensively explore children's profiles and the intricate relationships between their physical, psychological and social domains and their impact on CP.

## Materials and methods

### Participants

The required sample size to conduct the experiment was computed by using the G*Power 3.1 software (University of Düsseldorf, Düsseldorf, Germany). Based on a similar study investigating how student profiles based on motor competence, muscular fitness and cardiorespiratory fitness influence educational variables (e.g., mathematical skills) [34], it was determined that the effect size to be identified was Cohen's f = 0.272. Using this effect size, a Type I error

rate of 0.05, and a Type II error rate of 0.2, a sample size of 152 participants was obtained. Given that previous studies suggest relatively low compliance rates for wearing accelerometers long enough to ensure valid PA measurements (with dropping rates potentially exceeding 50%) [35–37], efforts were made to recruit double the minimum required sample size.

Before the study, the families and guardians received an informed consent form, which they were required to sign, and the participants volunteered to participate. The consent rate was 97.3%. The initial sample consisted of two hundred and ninety-five participants, 33.6% of whom were excluded for not meeting the accelerometry requirements (i.e. having, at least, valid data from three working days and one weekend day, with a minimum of 8 hours per day). Additionally, participants with physical or cognitive impairments that prevented them from performing the tests normally (n = 3) were allowed to complete the tests, but their data were excluded from the statistical analyses. Finally, data from one hundred ninety-four fifth-grade children (48.96% girls) with a mean age of 10.61 years (SD = 0.45) were employed in this cross-sectional study. Data collection for this study was conducted from on October 7, 2022, to March 24, 2023. All the tests, including the questionnaires and the physical and cognitive tests, were completed by the participants, who were from six schools located in the urban area of the province of Valencia (Valencian Community, Spain). The six participating schools were selected through randomization, with final inclusion based on their agreement and willingness to carry out the necessary assessments. The study procedures were conducted in accordance with the Helsinki Declaration, and the Ethical Committee of the University of Valencia approved the study (Code 1564606).

## Procedures

The study commenced by contacting the schools and their principals, explaining the research objectives and requesting their collaboration. After obtaining consent from the families, measurements were taken over the course of a single week in each school. During this week, data were collected on participants' daily PA levels using accelerometers, alongside written questionnaires, physical tests, and cognitive tests. Academic achievement data was obtained from the school principals who provided students' academic grades. The research team supervised all the tests.

The questionnaires were filled in in the children's regular classroom after an explanatory introduction. Questionnaires were projected onto a whiteboard, and a research assistant read each item aloud. Two additional research assistants supervised the entire process providing support to the participants when required. All the questions were answered during two 45-minute sessions held on different days within the same week. Following the questionnaires, the mathematical fluency test was administered to the class group and timed by the researchers. Cognitive tests were subsequently then conducted in a quiet environment on PCs under the supervision of the researchers. Finally, physical tests were conducted in two sessions, either in the school gym or schoolyard, during two PE lessons. Researchers provided explanations and demonstrations before each group of students performed them. All tests were managed by researchers.

## Measurements

**Daily physical activity.** Participants' daily movements (PA and sedentary behavior) were recorded by accelerometers (ActiGraph wGT3X-BT model), which have been shown to be reliable and valid for use in children [38]. The students were instructed in their use and wore them for 8 consecutive days. To ensure data validity, a criterion was set requiring at least of three working days and one weekend day with a minimum of 8 hours of accelerometer

recording per day. The 8 hours did not need to be registered continuously. The data was analyzed on Matlab R2022b software (MathWorks Inc., Natick, MA, USA).

The vertical axis accelerometer records were initially collected at 15-second intervals and then consolidated into 60-second periods. Segments with 20 or more consecutive minutes of 0 counts were identified, indicating the periods when the participants were not using the device. These data segments were excluded from the subsequent analysis. Thresholds were applied to categorize the minute-by-minute PA intensity [38]. Three intensity levels were defined: sedentary ($\leq$100 counts/min), light ($\leq$2295 counts/min) and moderate-to-vigorous ($>$2295 counts/min). The analysis accelerometer-derived metrics to calculate the average time spent in daily moderate-to-vigorous PA (MVPA).

**Self-reported physical, social and psychological factors.** Questions were selected from nine different questionnaires, while the input variables in the SOM were based on a prior analysis (manuscript under review). Table 1 gives the validation references, questionnaires used, and variables analyzed by each scale as regards the various factors related to the students' physical, social and psychological domains.

**Cognitive performance.** Cognitive data were collected to evaluate students' actual and perceived CP using objective and subjective instruments, respectively. Subjective CP was assessed through a self-report on language and mathematics competence, adapted from the NEPS matrix test [48] scored on a 5-point Likert scale. A mathematical fluency test [49] and two cognitive tests were administered to objectively measure the CP. The math test assessed

**Table 1. Details of the self-reported questionnaires used in the study according to the domain.**

| Reference | Questionnaire | Variable | Domain |
|---|---|---|---|
| Menescardi et al. [39] | Physical Activity Questionnaire for Children (PAQ-C). This questionnaire has demonstrated good validity [$\chi^2$ (20) = 33.583; CFI = 0.98; RMSEA = 0.04; SRMR = 0.03] and reliability ($\alpha$ = 0.73) in previous studies. | Self-reported PA | Physical |
| Estevan et al. [40] | Pictorial scale of Perceived Movement Skill Competence (PMSC). This questionnaire has demonstrated robust structural validity [$\chi^2$ = 29.50; df = 14, p < 0.01; RMSEA = 0.035; SRMR = 0.020; CFI = 0.989; TLI = 0.983] and reliability, including internal consistency ($\alpha$ > 0.70) and test-retest reliability (intraclass correlation coefficient > 0.70). | Perceived motor competence | Psycho-logical |
| Ortega-Benavent et al. [41] | Physical Literacy for Children Questionnaire (PL-C Quest). This questionnaire has proved good structural validity [$\chi^2$ (219) = 304.366; CFI = 0.92; RMSEA = 0.04; SRMR = 0.06], while its reliability has been shown to be strong with internal consistency values ($\alpha$ = 0.84; $\omega$ = 0.83) and test-retest reliability (r = .84). | Perceived physical literacy | |
| Menescardi et al. [39] | Behavioural Regulation Exercise Questionnaire (BREQ). The psychometric properties of this questionnaire indicate satisfactory validity [$\chi^2$ (38) = 61.489; CFI = 0.97; RMSEA = 0.04; SRMR = 0.04] and reliability, with Cronbach's alpha values ranging from 0.54 to 0.72. | Intrinsic motivation | |
| | | Identified motivation | |
| Sebire et al. [42] | Basic psychological needs satisfaction within a PE setting. Evidence supports the strong validity of this instrument [$\chi^2$ (99) = 255.75; CFI = 0.95; RMSEA = 0.052; SRMR = 0.04]. Reliability was acceptable, with Cronbach's alpha value of 0.82 for competence subscale. | BPN competence | |
| Estevan et al. [43] | The pictorial scale of Physical Self-Concept in Children (P-PSC-C). This questionnaire has demonstrated good structural validity [$\chi^2$ (9) = 29.8; CFI = 0.99; RMSEA = 0.055; SRMR = 0.046], acceptable internal consistency ($\alpha$ = 0.74) and test-retest reliability (ICC ranging from 0.47 to 0.77). | Physical self-concept | |
| Sebire et al. [42] | Basic psychological needs satisfaction within a PE setting. Reliability was acceptable for relatedness subscale, with Cronbach's alpha value of 0.81. | BPN relatedness | Social |
| Castillo et al. [44] | Intrinsic Satisfaction Classroom Scale (ISC). This questionnaire has proven to be both valid [$\chi^2$(13) = 66, p < 0.01; df = 5.09; RMSR = .03; GFI = 0.98; AGFI = 0.96; CFI = 0.98] and reliable ($\alpha$ = 0.8). | School enjoyment | |
| Bruner and Benson [45] | Social Identity Questionnaire for Physical Education/Sport (SIQS). Previous studies have demonstrated the validity [$\chi^2$ (18) = 43.52; CFI = 0.99; TLI = 0.98; RMSEA = 0.04; SRMR = 0.02] and reliability ($\omega$ = 0.89) of this questionnaire. | Social identity | |
| Castillo et al. [46]; Newton et al. [47] | Learning Climate Questionnaire. Validity [$\chi^2$ (341) = 711.5; CFI = 0.9; RMSEA = 0.054] and reliability ($\alpha$ = 0.87) of this questionnaire have been well-established in previous research. | Task-involving climate | |

the ability to perform 160 simple mathematical operations of addition, subtraction, and multiplication) as quickly as possible. The Digit Span test and the Stroop test were used as individual cognitive tests. The assessment of academic achievement was based on the final grades provided by the schools for the entire academic year. The consistent grading system used across all participating schools ensured a uniform scoring approach. The schools provided scores ranging from 0 (i.e., worst academic performance) to 5 (i.e., best academic performance) for each participant and subject. To obtain a comprehensive score for each participant, the average grade across the school subjects was computed. As trimester evaluations are not conducted in these schools, the final grades represent the cumulative performance across the full academic year.

**Digit span test.** An adapted computer version of the Digit Span test was used to evaluate working memory using the PEBL2 software (The Psychology Experiment Building Language) [50]. The participants were required to recall a sequence of numbers and repeat them in the same order. The digital version of the test displayed numbers on a laptop screen for 1 second, followed by a 1-second interval before the next number. The participants then repeated the entire sequence on the laptop. The sequence length started at three digits and increased progressively, with participants allowed two attempts to repeat the numbers. The test concluded when a participant either failed both attempts or reached the maximum length of 10 numbers. The test results comprised the total number of correct sequences and the longest correctly recalled number sequence (i.e., block width). The Digit Span test has shown moderately high reliability in children [51].

**Stroop test.** An adapted computer version of the Victoria Stroop test, employing the PEBL2 software [50], was used to evaluate various aspects of executive functions, including selective attention, cognitive inhibition, cognitive flexibility and information processing speed [52]. The test consisted of three stages, each with four rows of six items, including at least one item in red, blue, yellow and green per row. The participants were required to scan the items in the rows from left to right and indicate the color of each item on the laptop keyboard. The initial dot task displayed colored dots as the items, while the second, the neutral word task, presented common words displayed in color. The third, known as the color word task, involved identifying the colors that did not match their names. The variables obtained from this task included (a) the completion time per round, (b) total responses per round, (c) accuracy in identifying colors and words, and (d) correct identification of colors and words.

**Physical test.** Three key measures were used in the physical tests. First, the Canadian Agility Movement Skill Assessment (CAMSA) was employed. This test is based on a circuit-based field test to measure actual motor competence. This test combined product- and process-oriented measures of fundamental movement skills [53]. The second physical test used was the Progressive Aerobic Cardiovascular Endurance Run (PACER) proposed by Morrow et al. [54]. This test allows to measure cardiorespiratory fitness level. In third place, children's weight and height were assessed to calculate their body mass index (BMI) percentiles, adjusted for gender and age, following Kuczmarski et al. [55].

## Self-organizing maps analysis procedure

Participants were classified using the SOM process, an unsupervised artificial neural network that generates profiles based on input variable similarities. Daily MVPA, cardiorespiratory fitness, actual motor competence, BMI%, perceived motor competence, perceived physical literacy, intrinsic motivation, identified motivation, perceived PA participation, perceived competence in PE, perceived relatedness in PE, physical self-concept, school enjoyment, social identity and task-involving climate were included as input variables. Matlab R2022b software (MathWorks Inc., Natick, MA, USA) and the SOM toolbox (version 2.0 beta; Laboratory of

computer and information science, Helsinki University of Technology, Helsinki, Finland) for Matlab were used to compute the SOM analysis (S1 File).

The SOM analysis involved the following training process [56]: first, the construction of a neuron network per input variable, whose size was determined by the number of cases included in the data set (i.e., 194 participants) and the proportion of the two main eigenvalues of the data set. This resulted in a neural network consisting of 12 x 6 neurons (height x width). Values were then assigned to each neuron for all input variables in both random and linear initialization, while the values of the initially assigned neurons were modified according to sequential and batch training algorithms. A visual representation of the SOM training process is shown in Fig 1. This flowchart illustrates the key steps in constructing and training the neural network, as well as the iterative adjustments of the neuronal weights.

Various factors influence the adjustment of the neuronal weights during each iteration of the training phase. Initially, an input vector (i.e. a participant) is introduced to the network. The neurons within the lattice engage in a 'competition' by comparing the Euclidean distance of their weight vector with the values of the input vector. The goal is to minimize the Euclidean distance between the weight vector and the input vector, aligning the winning neuron with its associated

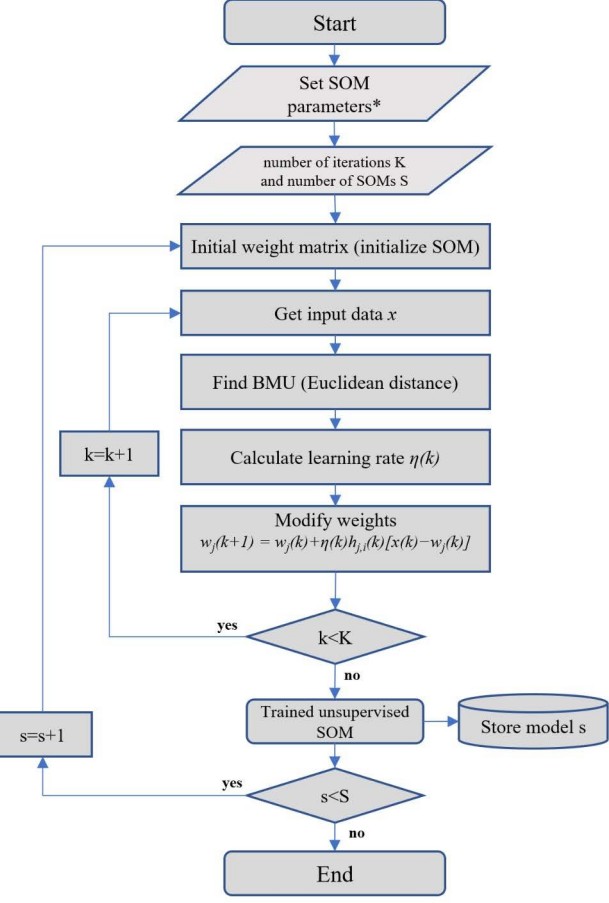

**Fig 1. Flowchart for the unsupervised SOM algorithm modified from Seifert et al. [56] with permission.** Here, $k$ denotes time, $x(k)$ is an input vector randomly drawn form the input data set at time $k$, $h_{j,i}$ denotes the neighborhood function around the winner unit $j$ and $\eta(k)$ the learning rate at time $k$.

input cases. All the neurons in the lattice then adjust their weight values to approach the values of the input vector. The extent of adaptation is influenced by two processes: the learning ratio and the neighbor function. The former is initially high and gradually diminishes throughout the training process. The latter dictates the adaptation of both the winning neuron and the surrounding neurons. The magnitude of adaptation is inversely linked to the distance between the neuron and the winner. This iterative process continues until the training process is completed.

The final analysis depends on a random procedure, including the initialization and entry order of the input vector. To find the best solution, the process was repeated 100 times. A total of 1600 SOMs were obtained from two different training methods, four neighborhood functions, and two initialization methods (i.e. $100 \times 2 \times 4 \times 2$). This number of iterations was selected as it provides a sufficient balance between computational efficiency and the level of convergence necessary for robust results, minimizing the risk of underfitting while allowing the map to capture meaningful patterns in the data. Stability was assessed by monitoring the consistency of the cluster configurations across iterations, ensuring that the profiles identified remained similar after repeated runs. The map with the minimum error was selected by calculating the product of quantization and topographic errors [57].

The k-means method was used to categorize neurons into broader groups based on the characteristics of the input variables. It is important to note that the input data in this case consisted of the neural weights derived from the SOM analysis and not the subjects' values. The number of clusters ranged from 2 to 10 to prevent an excessive number of profiles. The Davies–Bouldin index was used to evaluate the quality of the clustering configuration and to determine the optimal number of clusters. This index measures the average similarity ratio of each cluster with its nearest cluster, where lower values indicate greater distinctness and separation. In this study, the cluster configuration with the lowest Davies–Bouldin index was chosen, as it indicated the highest separation between profiles and minimized within-cluster variance, supporting the reliability of the selected clusters in representing distinct student profiles. These clusters were used to describe the characteristics of the physical, social, cognitive and psychological dimensions in the students analyzed.

To interpret the maps, all the subjects in a neuron must be considered to share certain characteristics (i.e. input variable values). Each participant (input case) is placed in a specific neuron according to its input variable values at the end of the SOM training process. The maps represent a network of neurons in which each neuron is a hexagon. Between the three dimensions shown, the x- and y-axes represent the topographical situation of each neuron, with the nearest neurons having similar weight values (i.e. input variable values) and the most distant neurons having the most different weight vectors. The remaining dimension is represented by the color of the map, which indicates the weight value of each neuron in the network for each specific input variable.

## Data analysis

The input variables from the physical, social, and psychological domains were used to profile the participants. Descriptive data on actual and perceived CP, math fluency, and academic achievement were then analyzed according to the identified profiles. First, the effect of clustering on the input variables was checked by the Kruskal‒Wallis test, with pairwise comparisons between clusters conducted using the Dunn–Bonferroni correction at a significance level of $p = 0.05$.

Secondly, the Kruskal‒Wallis test was applied to determine the cluster effect on both subjective (i.e. academic achievement, perceived mathematical performance and perceived language performance) and objective cognitive variables (i.e. math fluency, Digit Span [i.e. Total Recall Score] and Stroop variables [i.e. Dot Time, Word Time, Color Time, Correct Responses Dot, Correct Responses Word, Correct Responses Color, Dot Time Efficacy, Word

Time Efficacy, Dot Responses Efficacy and Word Responses Efficacy]). The Dunn-Bonferroni post-hoc correction was then applied.

Finally, gender was compared within the clusters using the chi-square test at a significance level of $p$ = 0.05 to identify any significant associations between cluster membership and gender.

## Results

This section provides a detailed description of the figures. Fig 2 shows the results of the SOM analysis. To facilitate its interpretation, an explanation is given below. Panels A, B, and C display the neuron maps for all the input variables, with each section representing one dimension. The maps show the distribution of the cases based on their characteristics (yellow [high values], blue [low values]).

The Davies–Bouldin index indicated a 6-cluster solution as the most appropriate (Fig 2D). The neuron distribution in each cluster is shown in Fig 2E. Different colors are assigned to each cluster. The size of the neuron's shadow represents the number of participants classified in each neuron, with larger shadows indicating more subjects.

Each component plane in Fig 2 represents the value of a variable for the participants assigned to each neuron. The weights remain in the same neuron in all the component planes. For example, the neuron in the lower right corner contains participants who have low scores on variables related to the physical dimension (i.e. daily MVPA, Cardiorespiratory fitness, Actual motor competence), high BMI percentile, low-to-medium scores on variables related to the psychological domain (i.e. perceived motor competence, perceived physical literacy, intrinsic motivation, identified motivation, perceived PA participation, BPN competence, and physical self-concept) and low to moderate scores on the variables in the social domain (i.e. BPN relatedness, school enjoyment, social identity and task-involving climate). The profiles established are described below.

Cluster 1 (Excellers) comprises participants with high values in all the domains (i.e. physical, psychological, social) and lower BMI percentiles than the rest of the clusters. Cluster 2 (Notables) is formed of children with inconsistent values in the physical domain (i.e. low values in MVPA and fitness, medium actual motor competence and self-reported PA, and a high BMI percentile). This group obtained a relatively high score in the variables relating to the psychological and social domains, with a solid rating on the variables perceived relatedness in PE and school enjoyment. This cluster contained the largest number of participants. The children allocated to cluster 3 (Resilients) presented low values in the physical domain, except for those obtained in the actual motor competence variable, which remained relatively high, together with a high BMI percentile. Moderate values were obtained for this group in the psychological and social domains, highlighting high intrinsic motivation and school enjoyment, together with a much lower perceived relatedness in PE than the other clusters. The participants in cluster 4 (Moderates) had medium-to-low scores in the physical domain with a tendency towards a low BMI. The scores were moderate in both the psychological and social domains in relation to those of the remaining clusters. Cluster 5 (Strugglers) showed a physical domain with low tendencies, except for the actual motor competence variable, in which moderate values were obtained. These participants' BMI percentile was relatively high. The psychological and social domains presented the lowest scores in the sample for all the variables, except for perceived motor competence, despite having obtained low values for this variable as well. This group had the second-highest number of participants. Finally, the participants in cluster 6 (Socializers) had very low scores in the physical domain and were overweight (90.6 BMI percentile). This group had low-to-moderate scores in the psychological domain, except for intrinsic

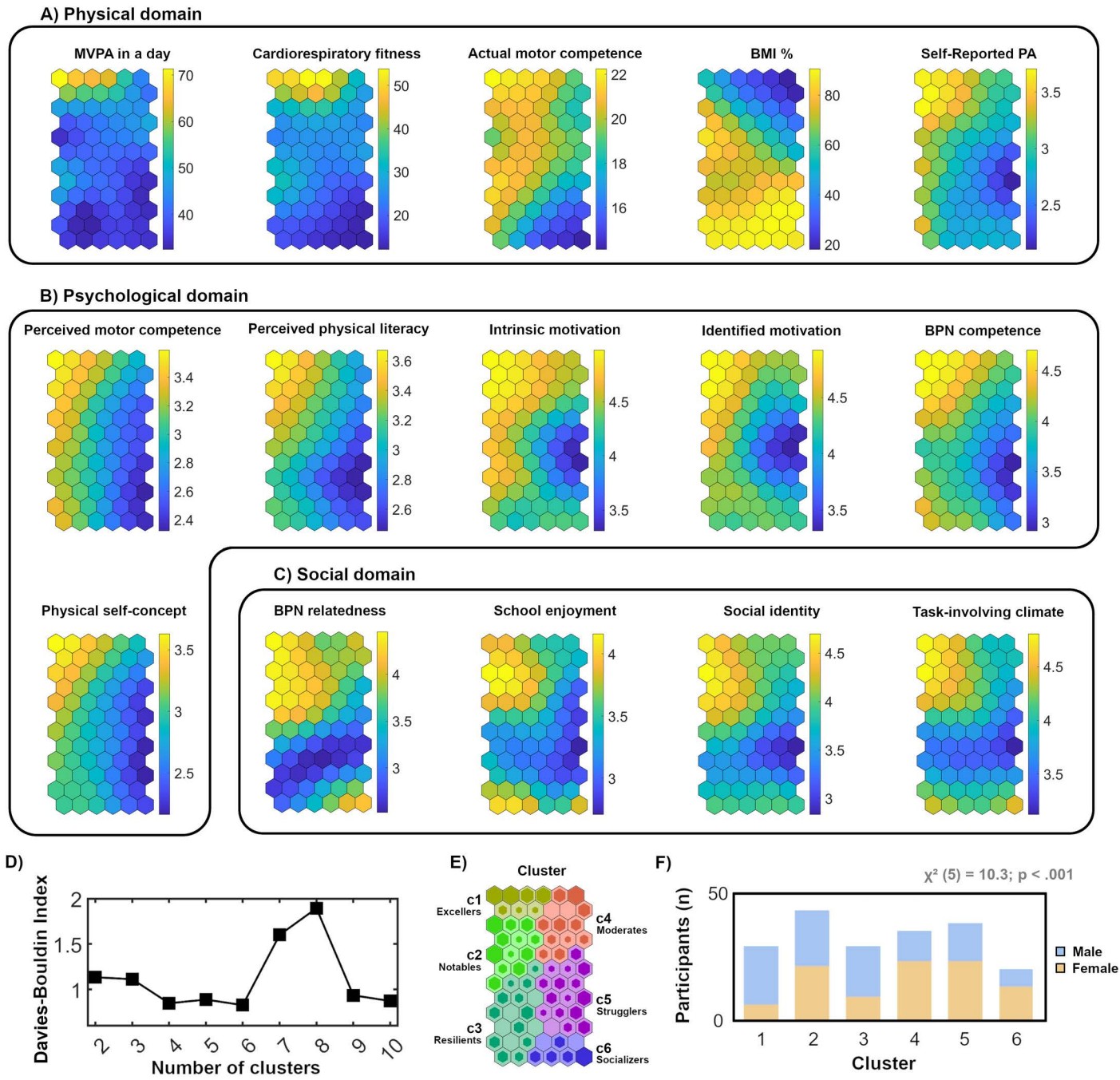

**Fig 2. Description of results of self-organizing map analysis.** The component planes from the analysis are shown in panels (A–C) for the physical, psychological, and social variables of the three domains, respectively. Panel (D) displays the Davies–Bouldin index related to the k-means cluster algorithm applied to set profiles. Panel (E) shows the obtained clusters and the hits (neuron shadows). Panel (F) shows the number of participants in each cluster, the girl/boy percentage in each group and the results of the chi-square test.

and identified motivation, which remained high. They also showed high scores in the social domain variables, excluding physical self-concept. This cluster had the smallest number of participants.

The comparison between the clusters (Kruskal-Wallis test) in terms of the SOM input variables showed a significant main effect of cluster membership (Table 2). The clusters' descriptive statistics and pairwise comparisons are shown in Table 3.

The comparison of the cognitive domain variables in the different clusters revealed a significant cluster effect on academic achievement ($H_5 = 18.83$; $p < 0.002$). However, no significant effects were observed for the other subjective cognitive variables (i.e. perceived mathematical performance and perceived language performance). Pairwise comparisons (Fig 3) revealed that clusters 1 and 4 had higher academic achievement scores than cluster 5.

Table 4 shows the cluster effect on objective cognitive variables math fluency, Digit Span test and Stroop test variables. The pairwise comparisons are shown in Table 5.

## Discussion

Examining the relationships among the physical, psychological and social domains primarily related to PA and their influence on CP enhances our understanding of human development, providing a basis for interventions aimed at fostering these areas during childhood. This study aimed to examine the combined impact of these factors on children's CP, by analyzing them interactively rather than in isolation, offering a holistic understanding of their interplay. This approach led to the identification of six distinct profiles, each associated with different CP levels. While previous research has often focused on isolated relationships —such as between physical and cognitive [3,10,12,14], physical and psychological [25–27], physical and social [6,30], cognitive and psychological [23,29], or cognitive and social [31,32]— the person-centered approach used here provides a more comprehensive view. This approach reveals nuanced profiles by considering the dynamic interactions among domains, thus enriching our understanding of individual CP development pathways.

This perspective allows for the observation of variable interactions within the generated profiles, avoiding the loss of relevant information in their interaction process. Overall, profiles with high scores across several domains (i.e. Excellers, Notables, Resilients and Moderates) generally exhibit moderate-to-high CP levels. In contrast, profiles with low scores in one or more domains (i.e. Strugglers and Socializers) usually exhibit poor scores

**Table 2. Cluster effect on input variables.**

| Variables | $\chi^2$ (5) | p | $\varepsilon^2$ |
|---|---|---|---|
| MVPA in a day | 42.3 | <0.001 | 0.22 |
| Cardiorespiratory fitness | 82.6 | <0.001 | 0.43 |
| Actual motor competence | 62.1 | <0.001 | 0.32 |
| BMI % | 97.5 | <0.001 | 0.51 |
| Self-Reported PA | 67.8 | <0.001 | 0.35 |
| Perceived motor competence | 115.0 | <0.001 | 0.60 |
| Perceived physical literacy | 105.4 | <0.001 | 0.55 |
| Intrinsic motivation | 95.2 | <0.001 | 0.49 |
| Identified motivation | 87.4 | <0.001 | 0.45 |
| Perceived competence in PE | 77.1 | <0.001 | 0.40 |
| Physical Self-Concept | 107.4 | <0.001 | 0.56 |
| Perceived Relatedness in PE | 84.0 | <0.001 | 0.44 |
| School enjoyment | 41.2 | <0.001 | 0.21 |
| Social identify | 75.3 | <0.001 | 0.39 |
| Task involving climate | 70.0 | <0.001 | 0.36 |

**Table 3. Descriptive statistics of input variables and pairwise comparisons.**

| | Cluster 1 Excellers n = 29 | Cluster 2 Notables n = 43 | Cluster 3 Resilients n = 29 | Cluster 4 Moderates n = 35 | Cluster 5 Strugglers n = 38 | Cluster 6 Socializers n = 20 |
|---|---|---|---|---|---|---|
| MVPA in a day | 70.43* (32) | 34.86[1] (22.2) | 34[1,4] (21.1) | 40.57[1,3,5] (22.3) | 33.57[1,4] (21.4) | 33.88[1] (20.5) |
| Cardiorespiratory fitness | 56* (20) | 26[1,5,6] (15.5) | 22[1,2,6] (15) | 28[1,5,6] (15) | 19[1,2,4,6] (11.8) | 11* (6.5) |
| Actual motor competence | 22[4,5,6] (4) | 21[4,5,6] (5) | 20[5,6] (3) | 19[1,2,6] (3.5) | 19[1,2,3,6] (4) | 14.5* (3.25) |
| BMI % | 37[2,3,5,6] (32.4) | 81.5[1,4] (22.5) | 79.5[1,4] (20.9) | 28.1[2,3,5,6] (25.7) | 79.35[1,4,6] (27.2) | 90.6[1,4,5] (8.6) |
| Self-Reported PA | 3.48[3,4,5,6] (0.49) | 3.42[3,4,5,6] (0.91) | 2.99[1,2,5] (0.66) | 3.02[1,2,5] (0.57) | 2.30* (0.89) | 2.89[1,2,5] (0.51) |
| Perceived motor competence | 3.55[3,4,5,6] (0.3) | 3.35[4,5,6] (0.45) | 3.15[1,4,5,6] (0.4) | 2.7[1,2,3] (0.63) | 2.58[1,2,3] (0.44) | 2.53[1,2,3] (0.63) |
| Perceived physical literacy | 3.63[3,4,5,6] (0.23) | 3.43[3,4,5,6] (0.28) | 3.17[1,2,5,6] (0.37) | 2.97[1,2,5] (0.33) | 2.6[1,2,3,4] (0.52) | 2.75[1,2,3] (0.48) |
| Intrinsic motivation | 5[3,4,5,6] (0) | 5[3,4,5,6] (0.33) | 4.67[1,2,5] (0.67) | 4.67[1,2,5] (0.33) | 3.67* (1) | 4.33[1,2,5] (0.75) |
| Identified motivation | 5[3,4,5,6] (0.33) | 5[3,4,5,6] (0.67) | 4.33[1,2,5] (0.67) | 4.33[1,2,5] (0.67) | 3.67* (0.33) | 4.33[1,2,5] (0.67) |
| BPN competence | 4.8[3,4,5,6] (0.4) | 4.4[4,5,6] (1) | 4.2[1,5] (0.6) | 4[1,2,5] (0.8) | 3.2* (0.8) | 3.8[1,2,5] (0.65) |
| Physical Self-Concept | 3.5* (0.5) | 3.33* (0.5) | 3[1,2,4,5,6] (0.33) | 2.67[1,2,3] (0.5) | 2.25[1,2,3,4] (0.63) | 2.67[1,2,3] (0.54) |
| BPN relatedness | 4.2[3,5] (0.8) | 4.6[3,4,5] (0.8) | 3[1,2,4,6] (1) | 4[2,3,5] (0.6) | 3[1,2,4,6] (1) | 4.1[3,5] (1.1) |
| School enjoyment | 3.8[5] (1) | 4[3,4,5] (0.8) | 3.6[2,5] (1) | 3.4[2,5] (0.9) | 2.8* (1) | 3.8[5] (1.2) |
| Social identify | 4.56[5] (1.22) | 4.56[3,4,5,6] (0.72) | 3.67[1,2,5] (0.89) | 4[2,5] (0.72) | 3.22* (0.64) | 3.89[2,5] (0.61) |
| Task-involving climate | 4.6[3,4,5] (0.8) | 4.6[3,4,5] (0.8) | 3.6[1,2,6] (0.8) | 4[1,2,5] (0.4) | 3.4[1,2,4,6] (0.6) | 4.4[3,5] (0.61) |

Data are expressed as median (interquartile range).

*Indicates significant differences with all clusters.

[1]Indicates significant differences with Cluster 1.

[2]Indicates significant differences with Cluster 2.

[3]Indicates significant differences with Cluster 3.

[4]Indicates significant differences with Cluster 4.

[5]Indicates significant differences with Cluster 5.

[6]Indicates significant differences with Cluster 6.

in CP outcomes. The initial analysis suggests a consistent interrelationship among the domains, as children with high values in the physical domain often demonstrate similarly high values in the psychological, social and cognitive domains, indicating a strong cross-domain synergy.

The profiles with generally high scores in the physical domain (i.e. Excellers and Moderates) showed elevated academic achievement scores, corroborating previous studies linking high cardiorespiratory fitness and lower BMI to better cognitive outcomes [58–60]. Within the physical domain, cardiorespiratory fitness [14–16] and MVPA [10,11,17] appear to play particularly critical roles, as these components are often associated with improved CP and

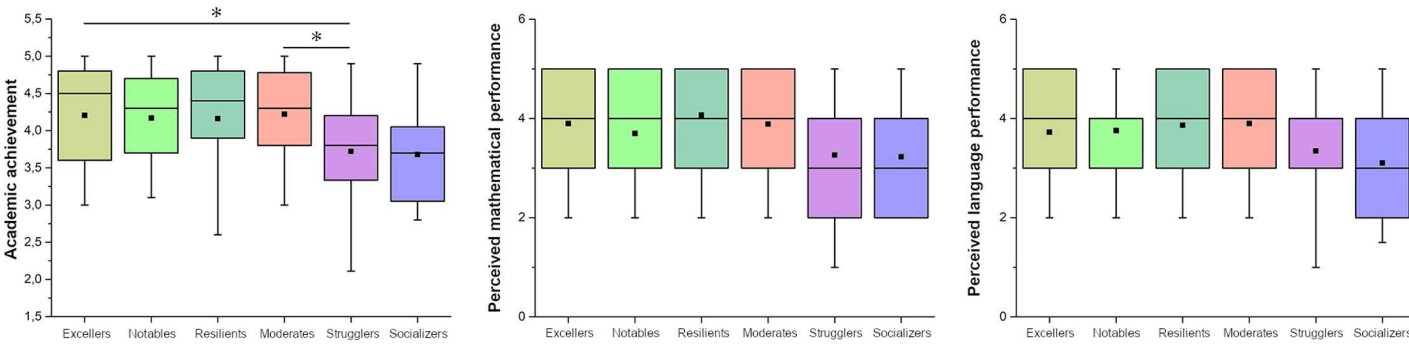

**Fig 3. Pairwise comparisons of cognitive domain variables between clusters.** The box plots display median values, interquartile ranges, and outliers for each cluster across three variables: Academic achievement (left), perceived mathematical performance (center), and perceived language performance (right). Significant differences between clusters are indicated by asterisks (*p < 0.05).

**Table 4. Cluster effect on digit span test and stroop test variables.**

| Variables | $\chi^2$ (5) | p | $\varepsilon^2$ |
|---|---|---|---|
| *Mathematical fluency* | | | |
| Correct operations | 28.54 | <0.001 | 0.1479 |
| *Digit span test* | | | |
| Total recall score | 4.43 | 0.49 | 0.02293 |
| *Stroop test* | | | |
| Dot time | 8.49 | 0.131 | 0.04398 |
| Word time | 17.67 | 0.003 | 0.09154 |
| Color time | 13.68 | 0.018 | 0.07086 |
| Correct responses dot | 16.54 | 0.005 | 0.08571 |
| Correct responses word | 4.98 | 0.418 | 0.02583 |
| Correct responses color | 7.69 | 0.174 | 0.03983 |
| Dot time efficacy | 5.6 | 0.347 | 0.02904 |
| Word time efficacy | 1.41 | 0.923 | 0.00732 |
| Dot responses efficacy | 2.86 | 0.722 | 0.01481 |
| Word responses efficacy | 4.12 | 0.532 | 0.02136 |

Dot time: The average time in seconds taken by the subject to complete the color in the dot task. Word time: The average time in seconds taken by the subject to complete the color in the neutral word task. Color time: The average time in seconds taken by the subject to complete the color in the color word task. Total responses dot: The average total responses made in the dot task. Total responses word: The average total responses made in the neutral word task. Total responses color: The average total responses made in the color word task. Dot time efficacy: The time used to complete the color word task divided by the time used to complete the dot task. Word time efficacy: The time used to complete the color word task divided by the time used to complete the neutral word task. Dot responses efficacy: The total responses used to complete the color word task divided by the total responses used to complete the dot task. Word responses efficacy: The total responses used to complete the color word task divided by the total responses used to complete the neutral word task.

academic achievement. This reinforces the importance of fostering opportunities for regular PA and sporting activities among younger pupils.

Conversely, profiles with moderate scores in the physical domain (i.e. Notables and Resilients) exhibited stronger CP in objective measures, such as math fluency and Stroop test performance, compared to profiles with generally low physical domain scores (i.e. Strugglers and Socializers). In their study, Katagiri et al. [61] identified the same relation.

Compensatory strengths in the psychological and social domains may enhance CP scores, offsetting moderate physical domain scores. Furthermore, the Strugglers and Socializers' profiles obtained the lowest CP scores. Both profiles exhibited low scores not only in the physical domain but also in the psychological domain. Notably, the Strugglers' profile also demonstrated low scores in the social domain. This observation underscores the importance of fostering development across all domains, as each one appears to play a complementary role in supporting cognitive outcomes. Additionally, as these low-CP profiles (i.e. Strugglers and Socializers) include a higher proportion of girls, ensuring equitable opportunities for both genders to engage in diverse forms of PA—aligned with their interests and beyond the traditional and stereotyped sports— becomes essential [62]. Promoting PA that caters to diverse motivations and preferences can help children maximize their physical and cognitive potential across domains. Encouraging varied forms of physical engagement may help bridge

**Table 5. Descriptive statistics of Digit Span test and Stroop test variables and pairwise comparisons.**

| Variables | Cluster 1 Excellers | Cluster 2 Notables | Cluster 3 Resilients | Cluster 4 Moderates | Cluster 5 Strugglers | Cluster 6 Socializers |
|---|---|---|---|---|---|---|
| *Mathematical fluency* | | | | | | |
| Correct operations | 26[c6] (6) | 29[c5,c6] (8.5) | 28[c5,c6] (7) | 23 (9.5) | 21[c2,c3] (11) | 21.5[c1,c2,c3] (8) |
| *Digit span test* | | | | | | |
| Total recall score | 5 (1) | 5 (1) | 5 (1) | 5 (1) | 4 (1) | 5 (1) |
| *Stroop test* | | | | | | |
| Dot time | 63.9 (25) | 64.5 (32.1) | 51.6 (36.3) | 64.3 (25.7) | 64.5 (30.9) | 76.1 (31.2) |
| Word time | 51.7 (19.8) | 52.1 (21.1) | 41.5[c4,c6] (13.5) | 53.5[c3] (30.3) | 48.5 (27) | 62.9[c3] (30.2) |
| Color time | 64.3 (19.6) | 59.5 (28.2) | 49.3[c6] (26.2) | 65.8 (31.2) | 58.7 (33.1) | 74.4[c3] (42.8) |
| Total responses dot | 24[c2] (1) | 25[c1] (2) | 25 (3) | 25 (3) | 24 (2) | 25.5 (1.25) |
| Total responses word | 24 (1) | 25 (2) | 25 (4) | 25 (2.5) | 25 (2) | 25 (2) |
| Total responses color | 25 (3) | 25 (3) | 26 (4) | 26 (4) | 25 (3) | 26.5 (1.25) |
| Dot time efficacy | 0.91 (0.72) | 0.94 (0.3) | 0.83 (0.34) | 0.97 (0.5) | 0.82 (0.43) | 1.06 (0.46) |
| Word time efficacy | 1.16 (0.55) | 1.08 (0.51) | 1.15 (0.33) | 1.1 (0.4) | 1.11 (0.33) | 1.15 (0.46) |
| Dot responses efficacy | 1.04 (0.16) | 1 (0.1) | 1 (0.09) | 1.04 (0.12) | 1 (0.12) | 1.04 (0.1) |
| Word responses efficacy | 1 (0.08) | 1 (0.12) | 1 (0.12) | 1 (0.16) | 1 (0.12) | 1.06 (0.08) |

Dot time: The average time in seconds taken by the subject to complete the color in the dot task. Word time: The average time in seconds taken by the subject to complete the color in the neutral word task. Color time: The average time in seconds taken by the subject to complete the color in the color word task. Total responses dot: The average total responses made in the dot task. Total responses word: The average total responses made in the neutral word task. Total responses color: The average total responses made in the color word task. Dot time efficacy: The time used to complete the color word task divided by the time used to complete the dot task. Word time efficacy: The time used to complete the color word task divided by the time used to complete the neutral word task. Dot responses efficacy: The total responses used to complete the color word task divided by the total responses used to complete the dot task. Word responses efficacy: The total responses used to complete the color word task divided by the total responses used to complete the neutral word task.

Significant gender differences were detected using the Chi-square test ($\chi^2_5$ = 21.0; $p < .001$). Clusters 1 and 3 had a greater number of boys, while clusters 4, 5, and 6 had a greater number of girls. Cluster 2 had a more balanced gender distribution.

the CP gap, particularly among children who may not naturally gravitate toward conventional PA.

It is crucial to highlight the cluster profiles that have consistently achieved higher results across various domains, as in the case of Excellers and Notables. The Excellers' profile exhibited considerable differences from the Notables, including higher scores in MVPA, cardiorespiratory fitness, and physical self-concept, as well as lower BMI. Given the similarities in these groups' psychological and social variables, it should be noted that their main distinctions lie within the physical domain. This observation suggests that providing targeted support in the physical domain for children within the Notables profile may help them transition closer to the Excellers profile, which demonstrated more favorable overall scores. Therefore, it may be beneficial to consider strategies that enhance these students' physical self-concept, daily PA, and cardiorespiratory fitness through Physical Education and extra-curricular sports activities. Progress in PA-related areas could positively contribute not only to their CP development [3–5] but also to improved physical self-perception as they mature [23]. In this regard, families and the educational community can play pivotal roles [6,30,31] with families promoting active lifestyles at home and schools providing structured opportunities to build fitness, confidence, and a positive physical self-concept, considering children's interests and motivations.

Among profiles with moderate scores across most domains, Resilients and Moderates displayed notable characteristics. The Resilients, predominantly male (69%), proved to be highly competent in objective CP tests, scoring significantly higher than Moderates, Strugglers and Socializers, despite having only moderate scores in the physical, psychological and social domains. In contrast, the Moderates, mostly female (65.7%), obtained the lowest BMI among all profiles and similar scores in psychological and social variables compared to Resilients. Moderates reported lower perceived motor competence and physical self-concept, but achieved higher scores in BPN relatedness. These findings suggest that while Moderates show lower perceived competence [63,64], their consistent engagement in PA could be influenced by higher levels of relatedness and social support. The observed differences between these two profiles may partially reflect gender-related disparities in physical development and the influence of socialization on physical competence perceptions [23]. Understanding these factors is crucial, as they can shape motivation and participation in PA, particularly as children transition into adolescence. To support the optimal development of students within these profiles, it is essential to implement targeted strategies that promote PA, develop a positive self-concept and enhance perceived motor competence. School-based interventions that promote diverse PA options [65] reduce screen time [66] could address the tendency of these profiles to undervalue their physical abilities, despite often exhibiting strong physical domain development. Such measures can foster these improvements, emphasizing the crucial roles of the educational community and family environment in promoting active, healthy lifestyles.

The Strugglers profile (60.05% girls) consistently scored low across most domains. Despite not being the least proficient physically, these individuals showed the lowest values in both psychological and social domains, resulting in poor CP in both objective and subjective measures. This suggests that the children within this profile may be at risk of atypical development, primarily due to low motivation and limited social support. Although the Strugglers scored higher in actual motor competence than Socializers, they displayed significantly lower scores in motivational aspects and overall social engagement. This highlights the need for targeted interventions that focus on enhancing motivation and fostering a supportive social environment, as deficits in these areas can negatively influence PA engagement and related psychological outcomes [67]. Tailored Physical Education programs that emphasize building motivation and strengthening social connections could play a crucial role in supporting these

students [3–5]. Educators should prioritize early identification and provide targeted support to address these challenges effectively and mitigate potential development risks.

The Socializers' profile (65% girls), displayed moderate-to-low scores in physical and psychological domains, as well as in CP, except for strong performance in the social domain, which occasionally exceeded that of Resilients and Strugglers. Despite their social strengths, the Socializers exhibited low levels of PA, actual motor competence and physical fitness, alongside high BMI and low self-perception. Nevertheless, their CP was the lowest across all profiles. Interestingly, their moderate scores in intrinsic and identified motivation, as well as their positive social self-perception and school enjoyment suggest that social support from family, teachers and peers may play a compensatory role, bolstering their engagement despite weaker physical performance [68]. In light of these findings, strategies that leverage their social skills and motivation for sports could enhance their physical, psychological, and cognitive development. Novel pedagogical strategies in Physical Education could promote extra-curricular engagement and improve their overall profile [69]. Future longitudinal studies are needed to explore whether such strategies could facilitate a shift towards profiles with higher physical, psychological and cognitive attributes.

It is important to point out that this study is not without limitations. As previously stated, it is crucial to consider the importance of examining different domains when creating profiles and their relationships with CP. Additionally, unmeasured variables, such as socio-economic level [70], nutrition [71], and sleep time [72], may significantly impact profile development and CP. The analysis relied on a cross-sectional cohort design, and future longitudinal studies are needed to track the changes in the variables over time. While the selected CP tests provided valuable insights, they revealed subtle differences among students within the same grade, suggesting the need for a broader range of tests to capture diverse executive functions. Expanding the assessment could help identify more distinct cognitive strengths and weaknesses across profiles, enabling the development of more targeted educational strategies.

In summary, this study used a person-centered approach with SOM to identify six distinct student profiles, shaped by varying interactions across physical, psychological and social domains. Our findings reveal that profiles with moderate-high scores in physical (i.e. Excellers, Notables, Resilients and Moderates) consistently obtained higher CP scores, as evidenced by superior results in objective cognitive measures (e.g. Stroop test and math fluency). In contrast, profiles with lower scores in physical and psychological domains (i.e. Strugglers and Socializers) exhibited the lowest CP, underscoring the critical role of these domains in cognitive development.

The analysis highlights the complex, non-linear evolution of these domains, revealing unexpected patterns, particularly in the Strugglers and Socializers, who did not conform to typical assumptions linking low physical competence to poor CP. Instead, motivational deficits and weaker social engagement appeared to contribute more significantly to their lower cognitive outcomes. These insights emphasize the need to look beyond PA promotion alone, focusing on enhancing self-perception, motivation, and social support as part of a comprehensive strategy for cognitive development.

Understanding these distinct profiles offers valuable guidance for educators and stakeholders in educational and sporting contexts. Tailored interventions addressing the specific strengths and weaknesses of each profile can reduce inequalities and foster inclusivity. By promoting PA and supporting positive self-concept development, such targeted programs can enhance children's physical, cognitive, psychological, and social well-being. Future research should prioritize longitudinal studies to monitor the evolution of these profiles and implement personalized interventions that align outcomes across all domains, ultimately improving educational and social conditions for all children.

## Supporting information

**S1 File. Matlab code developed to perform self-organizing maps analysis.**
(TXT)

## Acknowledgments

The authors would like to express their gratitude to the heads, students, parents and teachers of the participating schools for their cooperation.

## Author contributions

**Conceptualization:** Isaac Estevan.

**Data curation:** Sergio Montalt-García.

**Formal analysis:** Xavier García-Massó.

**Funding acquisition:** Isaac Estevan, Xavier García-Massó.

**Investigation:** Sergio Montalt-García.

**Methodology:** Sergio Montalt-García, Israel Villarrasa-Sapiña.

**Project administration:** Isaac Estevan.

**Resources:** Isaac Estevan, Israel Villarrasa-Sapiña, Xavier García-Massó.

**Software:** Xavier García-Massó.

**Supervision:** Isaac Estevan, Xavier García-Massó.

**Validation:** Israel Villarrasa-Sapiña.

**Visualization:** Sergio Montalt-García, Xavier García-Massó.

**Writing – original draft:** Sergio Montalt-García.

**Writing – review & editing:** Isaac Estevan, Israel Villarrasa-Sapiña, Xavier García-Massó.

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
