## [Decision Letter · Decision Letter 0]

28 Oct 2024

PONE-D-24-41678A person-centered approach to cognitive performance analysis in primary school children: comparisons through self-organizing mapsPLOS ONE

Dear Dr. García-Massó,

Thank you for submitting your manuscript to PLOS ONE. After careful consideration, we feel that it has merit but does not fully meet PLOS ONE’s publication criteria as it currently stands. Therefore, we invite you to submit a revised version of the manuscript that addresses the points raised during the review process.

We look forward to receiving your revised manuscript.

Kind regards,

Henri Tilga, PhD

Academic Editor

PLOS ONE

Journal requirements: When submitting your revision, we need you to address these additional requirements. 1. Please ensure that your manuscript meets PLOS ONE's style requirements, including those for file naming. The PLOS ONE style templates can be found at https://journals.plos.org/plosone/s/file?id=wjVg/PLOSOne_formatting_sample_main_body.pdf and https://journals.plos.org/plosone/s/file?id=ba62/PLOSOne_formatting_sample_title_authors_affiliations.pdf 2. Please include a caption for figure 1 and 2. 3. Please note that PLOS ONE has specific guidelines on code sharing for submissions in which author-generated code underpins the findings in the manuscript. In these cases, we expect all author-generated code to be made available without restrictions upon publication of the work. Please review our guidelines at https://journals.plos.org/plosone/s/materials-and-software-sharing#loc-sharing-code and ensure that your code is shared in a way that follows best practice and facilitates reproducibility and reuse. 4. We note that the grant information you provided in the ‘Funding Information’ and ‘Financial Disclosure’ sections do not match.  When you resubmit, please ensure that you provide the correct grant numbers for the awards you received for your study in the ‘Funding Information’ section.

Reviewers' comments:

Reviewer's Responses to Questions

**Comments to the Author**

1. Is the manuscript technically sound, and do the data support the conclusions?

Reviewer #1: Yes

2. Has the statistical analysis been performed appropriately and rigorously? 

Reviewer #1: Yes

3. Have the authors made all data underlying the findings in their manuscript fully available?

Reviewer #1: Yes

4. Is the manuscript presented in an intelligible fashion and written in standard English?

Reviewer #1: Yes

5. Review Comments to the Author

Reviewer #1: Greetings! I would like to express my appreciation for the excellent manuscript on this compelling topic. The depth of research and analysis is commendable.

I have reviewed the document and would like to offer a few suggestions that involve minor adjustments. I believe these revisions could further enhance the clarity and impact of your work.

Thank you for considering my feedback.

Abstract:

1. The aim of the study could be articulated more clearly. Instead of just saying "profile children," state explicitly why profiling is important (e.g., to understand how different domains affect cognitive performance).

If possible, rewrite it along the lines of “The objective of this study was to identify distinct student profiles based on physical, psychological, and social characteristics, and examine their impact on cognitive performance.”

2. Briefly add what Self-Organizing Maps (SOMs) are or their relevance to non-expert readers. A single sentence from the methodology can be added here “an unsupervised neural network technique for clustering”.

3. The mention of "six profiles" is vague. The abstract does not highlight any specific patterns or practical insights from the profiles. Consider mentioning key differences (e.g., “Profiles with higher physical activity levels were associated with better cognitive performance”).

4. The conclusion suggests that addressing student profiles can reduce inequalities and foster inclusiveness, which may be too broad without supporting results in the abstract. And as the study was conducted in only selected schools of a selected province, wording should ensure avoidance of such generalizable claims.

Introduction:

1. The introduction is quite dense and could benefit from better structuring. Several factors from physical, psychological, and social domains are listed without clear transitions, making it hard to follow the narrative. The contents can be shortened and reorganized under three subtopics: Physical Factors, Psychological Factors, and Social Factors to enhance readability, with much alteration.

2. Some statements are repeated or too similar (e.g., the positive correlation between physical activity and CP appears multiple times). These can be clustered together.

Methodology:

1. Please include what was the originally calculated sample size and how was it calculated?

2. Please mention how and why the six schools were chosen.

3. Clarify the specific requirements for accelerometry that led to the exclusion of 33.6% of participants.

4. Group the sentences regarding students volunteering and consent was taken from them and their parents, with the information regarding consent being first.

5. Explain why participants with physical or cognitive impairments were tested but excluded from analysis (Lines 149-151).

6. Specify how non-wearing segments of 20 minutes or more were handled (e.g., were these segments excluded, or was interpolation used?) (Line 185).

7. Please state if valid days required continuous wear for 8 hours or if gaps were permitted (Line 179).

8. Indicate if teachers or school staff were involved in conducting the tests, or if it was entirely managed by researchers (Lines 170-172).

9. Clarify whether the Digit Span and Stroop test versions used were standard or modified.

10. Mention a bit more regarding the type of academic grades obtained, whether these schools had the same grading systems or not. Furthermore, clarify the time frame utilized for evaluating academic achievement, specifying whether the assessment was based on grades from the past six months or from the entire academic year or more.

11. Explain the importance of 100 iterations in achieving reliable results and how stability was assessed. Also add a little on the Davies–Bouldin index to explain why the selected cluster configuration is optimal.

Discussion:

There is redundancy when explaining the interplay between physical, psychological, social, and cognitive domains across profiles. Streamline repetitive sections to make the text more concise without losing essential meaning.

Some profiles (e.g., Excellers and Strugglers) are discussed at length, while others (e.g., Socializers) are given less attention. Some findings, such as the differences between profiles (e.g., Resilients vs. Moderates), are not clearly summarized. Readers may struggle to identify the core insights. Ensure all critical claims are backed by citations or relevant studies.

6. PLOS authors have the option to publish the peer review history of their article (what does this mean? ). If published, this will include your full peer review and any attached files.

**Do you want your identity to be public for this peer review?** For information about this choice, including consent withdrawal, please see our Privacy Policy .

Reviewer #1: No

---

## [Author Response · Author response to Decision Letter 0]

3 Dec 2024

Review Comments to the Author

Reviewer #1: Greetings! I would like to express my appreciation for the excellent manuscript on this compelling topic. The depth of research and analysis is commendable. I have reviewed the document and would like to offer a few suggestions that involve minor adjustments. I believe these revisions could further enhance the clarity and impact of your work. Thank you for considering my feedback.

AR: Thank you very much for your thoughtful and positive feedback on our manuscript. We are grateful for your kind words and for recognizing the value of our research. We also appreciate the constructive suggestions you provided. We believe that these adjustments will indeed enhance the clarity and impact of the paper, and we have implemented each of your recommendations accordingly.

Thank you once again for your time and valuable input.

Abstract:

1. The aim of the study could be articulated more clearly. Instead of just saying "profile children," state explicitly why profiling is important (e.g., to understand how different domains affect cognitive performance). If possible, rewrite it along the lines of “The objective of this study was to identify distinct student profiles based on physical, psychological, and social characteristics, and examine their impact on cognitive performance.”

AR: We appreciate your comment. We have added the suggested sentence as we feel it is perfectly in line with the primary objective of this manuscript. We believe that this change provides a better explanation of the aim of the study.

2. Briefly add what Self-Organizing Maps (SOMs) are or their relevance to non-expert readers. A single sentence from the methodology can be added here “an unsupervised neural network technique for clustering”.

AR: We are grateful for your feedback. The suggested sentence has been incorporated into the abstract to provide clarification on the specifics of this analytical approach. We believe that your proposal offers valuable insights that enhance the comprehension of our manuscript.

3. The mention of "six profiles" is vague. The abstract does not highlight any specific patterns or practical insights from the profiles. Consider mentioning key differences (e.g., “Profiles with higher physical activity levels were associated with better cognitive performance”).

AR: We acknowledge and value your commentary and concur with your viewpoint. The information presented in the abstract has been adapted by the addition of the following sentence:

“Profiles characterized by higher physical activity levels and positive social and psychological factors were associated with better cognitive performance, in contrast to profiles with lower levels in these domains.”

We believe that this has enhanced the text and provided a more suitable contribution to the existing research findings.

4. The conclusion suggests that addressing student profiles can reduce inequalities and foster inclusiveness, which may be too broad without supporting results in the abstract. And as the study was conducted in only selected schools of a selected province, wording should ensure avoidance of such generalizable claims.

AR: In order to align the conclusion with the proposed adaptation, the text has been modified and the following sentence has been added at the end of the abstract:

“The recognition of the diversity of student profiles in specific educational settings may facilitate the design of more targeted programs that address individual needs and strengths, thereby enhancing their development in these domains within similar educational contexts.”

In light of your input, we have endeavored to refrain from generalizing the results, ensuring that the information presented is limited to that outlined in the manuscript and the specifics of the study itself. We consider this to be a valuable suggestion that enhances the conceptualization and contextualization of our research, for which we are grateful for your contribution.

Introduction:

1. The introduction is quite dense and could benefit from better structuring. Several factors from physical, psychological, and social domains are listed without clear transitions, making it hard to follow the narrative. The contents can be shortened and reorganized under three subtopics: Physical Factors, Psychological Factors, and Social Factors to enhance readability, with much alteration.

AR: We sincerely appreciate this valuable comment and understand your request fully. In response, we have restructured the introduction to enhance coherence and readability. Following your suggestion, we have organized the content under the three proposed subtopics: Physical Factors, Psychological Factors, and Social Factors. Additionally, we have provided clearer transitions between paragraphs, aiming to create a more cohesive and narratively structured text. We hope these modifications enhance the quality of our work and facilitate a straightforward understanding of our arguments. Thank you again for this insightful feedback.

2. Some statements are repeated or too similar (e.g., the positive correlation between physical activity and CP appears multiple times). These can be clustered together.

AR: In response to your recommendation, we have consolidated repetitive statements and rephrased sections to deliver a more concise and coherent presentation of our information. We hope these changes improve the clarity and overall quality of our introduction, as well as the manuscript as a whole. Thank you very much for your helpful suggestion.

Methodology:

1. Please include what was the originally calculated sample size and how was it calculated?

AR: We have included a paragraph in the “participants” section in which it explained how the required sample size was calculated.

2. Please mention how and why the six schools were chosen.

AR: Thank you for this suggestion. The six participating schools were selected through randomization, with final inclusion based on their agreement and willingness to carry out the necessary assessments. This information has been added to the manuscript to clarify the selection process.

3. Clarify the specific requirements for accelerometry that led to the exclusion of 33.6% of participants.

AR: We are grateful for your feedback. This information is reflected in the section entitled "Measurements" and located under the subheading "Daily physical activity". However, in order to clarify in this section that 33.6% of participants were excluded from the study, an explanatory sentence has been added to the same paragraph on “Participants”:

“The initial sample consisted of two hundred and ninety-two participants, 33.6% of whom were eliminated for not meet the requirements for accelerometry (i.e. having, at least, valid data from three working days and one weekend day, with a minimum of 8 hours per day).”

4. Group the sentences regarding students volunteering and consent was taken from them and their parents, with the information regarding consent being first.

AR: In response to your request, the various sentences have been grouped together with the intention of providing coherence and cohesion to the section on participants. Accordingly, the information regarding consent has been placed prior to the voluntary nature of participation in the study. The aforementioned information has been situated at the commencement of the section dedicated to “Participants”, as follows (Lines 148-150):

“Before the study, the families and guardians were provided with an informed consent form, which they were required to sign, and the participants volunteered to participate. The consent rate was 97.3%.”

5. Explain why participants with physical or cognitive impairments were tested but excluded from analysis (Lines 149-151).

AR: We are grateful for your observation. Participants with physical or cognitive impairments were initially tested to ensure inclusivity in the data collection process and to gain a full understanding of the diverse student population. However, they were excluded from the final analysis to maintain methodological consistency, as the primary focus of the study was on general educational interventions that were not specifically tailored to address significant physical or cognitive disabilities. This exclusion allowed for a clearer interpretation of the results within the target population for this study. A total of three participants were excluded from the study.

6. Specify how non-wearing segments of 20 minutes or more were handled (e.g., were these segments excluded, or was interpolation used?) (Line 185).

AR: These segments were excluded. We have included this information in the manuscript (Line 195).

7. Please state if valid days required continuous wear for 8 hours or if gaps were permitted (Line 179).

AR: No, it is not necessary to register the 8 hours uninterruptedly. We have included this in the manuscript (Lines 189-190).

8. Indicate if teachers or school staff were involved in conducting the tests, or if it was entirely managed by researchers (Lines 170-172).

AR: We appreciate your comment. We have clarified the information by adding the following sentence: " All tests were managed by researchers ". We hope this clarifies the issue you have raised.

9. Clarify whether the Digit Span and Stroop test versions used were standard or modified.

AR: An adapted computer version of both tests was conducted using the PEBL (Psychology Experiment Building Language) program, as provided by Mueller & Piper (2014). The program version used was PEBL 2.1 (2018). The manuscript has been updated to include clarifications regarding the version types of both tests. We extend our gratitude for your valuable input.

- Mueller, S. T., & Piper, B. J. (2014). The psychology experiment building language (PEBL) and PEBL test battery. Journal of neuroscience methods, 222, 250-259.

- The PEBL Project. (2018). PEBL: The Psychology Experiment Building Language. Retrieved from http://pebl.sourceforge.net

10. Mention a bit more regarding the type of academic grades obtained, whether these schools had the same grading systems or not. Furthermore, clarify the time frame utilized for evaluating academic achievement, specifying whether the assessment was based on grades from the past six months or from the entire academic year or more.

AR: We appreciate your comment. We calculated an overall average grade that included all subjects, providing a comprehensive academic achievement score on a scale from 0 to 5 for comparability purposes. All schools involved in the study utilized the same grading system, which facilitated this uniform approach to scoring. Academic achievement was assessed based on final grades from the end of the academic year, as these schools do not conduct trimester evaluations. Thus, the assessment reflects performance across the entire academic year. This information has been updated in the manuscript to clarify the type of grades obtained, the consistency in grading systems across schools, and the time frame used for evaluating academic achievement. The following text has been updated (Line 215-222):

“The assessment of academic achievement was based on the final grades provided by the schools for the entire academic year. The consistent grading system used across all participating schools enabled a uniform scoring approach. The schools provided a score ranging between 0 (i.e., worst academic performance) and 5 (i.e., best academic performance) for each participant and subject. To obtain a unique comprehensive value for each participant the average grade of all the school subjects was calculated. As trimester evaluations are not conducted in these schools, the final grades represent the cumulative performance across the full academic year.”

11. Explain the importance of 100 iterations in achieving reliable results and how stability was assessed. Also add a little on the Davies–Bouldin index to explain why the selected cluster configuration is optimal.

AR: Thank you for your comment, the next information has been added to the manuscript in order to facilitate the interpretation of the analysis:

- Line 289-294: This number of iterations was selected as it provides a sufficient balance between computational efficiency and the level of convergence necessary for robust results, minimizing the risk of underfitting while allowing the map to capture meaningful patterns in the data. Stability was assessed by monitoring the consistency of the cluster configurations across iterations, ensuring that the profiles identified remained similar after repeated runs.

- Line 299-306: The Davies–Bouldin index was used to evaluate the quality of the clustering configuration and to determine the optimal number of clusters. This index measures the average similarity ratio of each cluster with the cluster most similar to it, where lower values indicate more distinct and well-separated clusters. In this study, the cluster configuration with the lowest Davies–Bouldin index was chosen, as it indicated the highest separation between profiles and minimized within-cluster variance, supporting the reliability of the selected clusters in representing distinct student profiles.

It is our contention that these contributions enhance the quality of our manuscript and facilitate its interpretation and understanding.

Discussion:

There is redundancy when explaining the interplay between physical, psychological, social, and cognitive domains across profiles. Streamline repetitive sections to make the text more concise without losing essential meaning. Some profiles (e.g., Excellers and Strugglers) are discussed at length, while others (e.g., Socializers) are given less attention. Some findings, such as the differences between profiles (e.g., Resilients vs. Moderates), are not clearly summarized. Readers may struggle to identify the core insights. Ensure all critical claims are backed by citations or relevant studies.

AR: We would like to express our gratitude for your valuable feedback. We have conducted a detailed examination of the observations pertaining to the redundancy in explaining the interrelationship between the physical, psychological, social, and cognitive domains across profiles. In response, the sections in question have been streamlined with a view to reducing repetition while preserving the essential meaning and insights of the analysis. The revised discussion now places greater emphasis on the most pertinent interactions and differences across domains, while also eliminating superfluous duplication.

Additionally, we have addressed the imbalance in the depth of discussion between profiles. In the updated manuscript, we ensured that all profiles, including Socializers, received adequate attention and that their characteristics were clearly described in relation to their cognitive performance. We also enhanced the comparison between Resilients and Moderates, providing a clearer summary of their distinctions and offering a better interpretation of their unique features and cognitive outcomes.

To further enhance the clarity and coherence of the manuscript, we have merged the final paragraph of the discussion with the conclusions. This adjustment helped to eliminate redundancy and provided a more integrated and concise summary of our findings, aligning the closing remarks with the main objectives of the study. Furthermore, we have ensured that all critical claims are now supported by appropriate citations and relevant studies.

We appreciate your constructive feedback and believe that these revisions have significantly strengthened the quality and impact of our study.

---

## [Decision Letter · Decision Letter 1]

29 Dec 2024

PONE-D-24-41678R1A person-centered approach to cognitive performance analysis in primary school children: comparisons through self-organizing mapsPLOS ONE

Dear Dr. García-Massó,

Thank you for submitting your manuscript to PLOS ONE. After careful consideration, we feel that it has merit but does not fully meet PLOS ONE’s publication criteria as it currently stands. Therefore, we invite you to submit a revised version of the manuscript that addresses the points raised during the review process.

We look forward to receiving your revised manuscript.

Kind regards,

Henri Tilga, PhD

Academic Editor

PLOS ONE

Journal Requirements:

Reviewers' comments:

Reviewer's Responses to Questions

**Comments to the Author**

1. If the authors have adequately addressed your comments raised in a previous round of review and you feel that this manuscript is now acceptable for publication, you may indicate that here to bypass the “Comments to the Author” section, enter your conflict of interest statement in the “Confidential to Editor” section, and submit your "Accept" recommendation.

Reviewer #2: All comments have been addressed

2. Is the manuscript technically sound, and do the data support the conclusions?

Reviewer #2: Yes

3. Has the statistical analysis been performed appropriately and rigorously? 

Reviewer #2: Yes

4. Have the authors made all data underlying the findings in their manuscript fully available?

Reviewer #2: Yes

5. Is the manuscript presented in an intelligible fashion and written in standard English?

Reviewer #2: No

6. Review Comments to the Author

Reviewer #2: in terms of references, please check the journal guidelines, authors have names and numbers in the text, so please correct it according to journal standards;

in line 168, please describe which specific measurements were performed;

please provide the values of validity and reliability of the questionnaires used for the children;

would it be possible for any kind of graphical representation of the Self-Organizing Maps? its quite difficult to understand its assessments and how it was applied...

7. PLOS authors have the option to publish the peer review history of their article (what does this mean? ). If published, this will include your full peer review and any attached files.

**Do you want your identity to be public for this peer review?** For information about this choice, including consent withdrawal, please see our Privacy Policy .

Reviewer #2: No

---

## [Author Response · Author response to Decision Letter 1]

20 Jan 2025

Review Comments to the Author

5. Is the manuscript presented in an intelligible fashion and written in standard English?

Reviewer #2: No

AR: Thank you for your observation. Although no specific errors were mentioned, we conducted a thorough review of the manuscript, focusing on clarity, grammatical accuracy, and adherence to academic English standards. As part of this process, we carefully identified and corrected any grammatical or typographical errors to ensure the text is clear and unambiguous. Additionally, we refined sentence structure, improved terminology, and addressed potential redundancies to enhance overall readability. We believe these revisions have significantly improved the manuscript and hope they adequately address your concerns.

Reviewer #2:

1. In terms of references, please check the journal guidelines, authors have names and numbers in the text, so please correct it according to journal standards;

AR: We appreciate your observation regarding the referencing style in our manuscript. We have carefully reviewed both the references in the text and the journal's guidelines to ensure compliance. According to the journal's instructions, the Vancouver citation style is used, which allows for numbered citations in square brackets. The guidelines explicitly state the following regarding in-text citations: "In the text, cite the reference number in square brackets (e.g., ‘We used the techniques developed by our colleagues [19] to analyze the data’)."

Based on this information and the regulations stipulated by the Vancouver citation style, we confirm that the citation format we employed—combining author names and citation numbers within the text—is correct (e.g., “O’Malley et al. [31] identified…”). We have thoroughly verified that our manuscript adheres to this format throughout.

If there are specific instances where the referencing style seems unclear or inconsistent, we would greatly appreciate it if you could point them out so that we can address them. Otherwise, we believe the current referencing format aligns with the journal's requirements.

2. In line 168, please describe which specific measurements were performed;

AR: In order to specify the measurements that were carried out during the aforementioned week of measurements, the subsequent sentence in which the measurements were directly mentioned without reference to the specific time of the measurements has been modified. Consequently, the following sentence has been revised and incorporated to clarify this aspect:

Line 164-166: “During this week, data were collected on participants' daily PA levels using accelerometers, alongside written questionnaires, physical tests, and cognitive tests.”

3. Please provide the values of validity and reliability of the questionnaires used for the children;

AR: We appreciate your suggestion. Following your proposal, we have thoroughly revised the manuscript and included the validity and reliability values of the questionnaires. These values have been added to the Table 1 (line 202), where they are mentioned, to ensure that the psychometric properties of each instrument are clearly detailed. We hope that these additions address your concerns and improve the clarity and rigor of the manuscript.

4. Would it be possible for any kind of graphical representation of the Self-Organizing Maps? it’s quite difficult to understand its assessments and how it was applied...

AR: Thank you for your valuable suggestion. We agree that a graphical representation can improve the understanding of the Self-Organizing Maps process and its application. To address this, we have included Figure 1 in the revised manuscript (line 272). This flowchart illustrates the key steps in constructing and training the neural network, as well as the iterative adjustments of the neuronal weights. We believe this addition provides clarity and enhances the accessibility of the methodology for readers.

---

## [Decision Letter · Decision Letter 2]

23 Jan 2025

A person-centered approach to cognitive performance analysis in primary school children: comparisons through self-organizing maps

PONE-D-24-41678R2

Dear Dr. García-Massó,

We’re pleased to inform you that your manuscript has been judged scientifically suitable for publication and will be formally accepted for publication once it meets all outstanding technical requirements.

Kind regards,

Henri Tilga, PhD

Academic Editor

PLOS ONE

Additional Editor Comments (optional):

Reviewers' comments:

Reviewer's Responses to Questions

**Comments to the Author**

1. If the authors have adequately addressed your comments raised in a previous round of review and you feel that this manuscript is now acceptable for publication, you may indicate that here to bypass the “Comments to the Author” section, enter your conflict of interest statement in the “Confidential to Editor” section, and submit your "Accept" recommendation.

Reviewer #2: All comments have been addressed

2. Is the manuscript technically sound, and do the data support the conclusions?

Reviewer #2: Yes

3. Has the statistical analysis been performed appropriately and rigorously? 

Reviewer #2: Yes

4. Have the authors made all data underlying the findings in their manuscript fully available?

Reviewer #2: Yes

5. Is the manuscript presented in an intelligible fashion and written in standard English?

Reviewer #2: Yes

6. Review Comments to the Author

Reviewer #2: all the changes requested by the reviewer were adressed and the paper, in my opinion, is now suitable for publication

7. PLOS authors have the option to publish the peer review history of their article (what does this mean? ). If published, this will include your full peer review and any attached files.

**Do you want your identity to be public for this peer review?** For information about this choice, including consent withdrawal, please see our Privacy Policy .

Reviewer #2: No

---

## [Editor Report · Acceptance letter]

PONE-D-24-41678R2

PLOS ONE

Dear Dr. García-Massó,

I'm pleased to inform you that your manuscript has been deemed suitable for publication in PLOS ONE. Congratulations! Your manuscript is now being handed over to our production team.

Kind regards,

on behalf of

Dr. Henri Tilga

Academic Editor

PLOS ONE